# A Physiologically Based Pharmacokinetic Model to Predict Determinants of Variability in Epirubicin Exposure and Tissue Distribution

**DOI:** 10.3390/pharmaceutics15041222

**Published:** 2023-04-12

**Authors:** Radwan Ansaar, Robyn Meech, Andrew Rowland

**Affiliations:** College of Medicine and Public Health, Flinders University, Adelaide, SA 5042, Australia

**Keywords:** PBPK, epirubicin, pharmacokinetics, drug exposure

## Abstract

Background: Epirubicin is an anthracycline antineoplastic drug that is primarily used in combination therapies for the treatment of breast, gastric, lung and ovarian cancers and lymphomas. Epirubicin is administered intravenously (IV) over 3 to 5 min once every 21 days with dosing based on body surface area (BSA; mg/m^2^). Despite accounting for BSA, marked inter-subject variability in circulating epirubicin plasma concentration has been reported. Methods: In vitro experiments were conducted to determine the kinetics of epirubicin glucuronidation by human liver microsomes in the presence and absence of validated UGT2B7 inhibitors. A full physiologically based pharmacokinetic model was built and validated using Simcyp^®^ (version 19.1, Certara, Princeton, NJ, USA). The model was used to simulate epirubicin exposure in 2000 Sim-Cancer subjects over 158 h following a single intravenous dose of epirubicin. A multivariable linear regression model was built using simulated demographic and enzyme abundance data to determine the key drivers of variability in systemic epirubicin exposure. Results: Multivariable linear regression modelling demonstrated that variability in simulated systemic epirubicin exposure following intravenous injection was primarily driven by differences in hepatic and renal UGT2B7 expression, plasma albumin concentration, age, BSA, GFR, haematocrit and sex. By accounting for these factors, it was possible to explain 87% of the variability in epirubicin in a simulated cohort of 2000 oncology patients. Conclusions: The present study describes the development and evaluation of a full-body PBPK model to assess systemic and individual organ exposure to epirubicin. Variability in epirubicin exposure was primarily driven by hepatic and renal UGT2B7 expression, plasma albumin concentration, age, BSA, GFR, haematocrit and sex.

## 1. Introduction

Epirubicin is member of the anthracycline class of antineoplastic drugs. Anthracyclines are among the most broadly effective classes of antineoplastic drugs, and epirubicin is among the most clinically important drugs in this class. Epirubicin is primarily used in combination therapies for the treatment of breast, gastric, lung and ovarian cancers and lymphomas [1]. Epirubicin has emerged as the preferred agent in this class due to the favourable cardiotoxicity profile and similar anti-tumour activity compared to other anthracyclines [1,2].

Epirubicin is administered intravenously (IV) over 3 to 5 min once every 21 days with dosing based on body surface area (BSA; mg/m^2^). Despite accounting for BSA, marked inter-subject variability in circulating epirubicin plasma concentration has been reported. Eksborg [3] reported 10-fold between-subject variability in the area under the plasma concentration time curve (AUC) for epirubicin despite normalising for dose and BSA. As with most antineoplastic drugs, epirubicin has a narrow therapeutic window whereby small differences in exposure can result in marked differences in treatment efficacy and tolerability [4]. These factors underpin the value of better understanding the physiological and environmental covariates influencing epirubicin exposure, particularly those that can direct a more appropriate initial dose selection. Current initial dose selection for epirubicin based on BSA alone routinely overestimates dose requirement [5] necessitating dose reductions in subsequent cycles due to cardiac and haematologic toxicities [6]. Dose reductions and interruptions are most commonly due to reductions in neutrophil and platelet count [7]. Grade 3–4 neutropenia occurs in 8.4–54.2% of patients receiving epirubicin and cyclophosphamide (EC) (90/600 mg/m^2^) treatment; thus, haematological toxicity is monitored and dose reductions are implemented between cycles on a case-by-case basis [8]. Therapeutic drug monitoring (TDM) may be utilised to guide epirubicin dosing, provided an exposure profile has been developed and a known target therapeutic window is established. Development of PK/PD models can be successfully deployed to inform TDM; however, the infrequent dosing schedule of epirubicin and relatively short terminal half-life (18–45 h) limit practicality in this setting [9,10]. Additionally, while TDM is appropriate to guide on-treatment dose adjustments for antineoplastic drugs [11,12], it does not support optimal initial (cycle 1) dose selection.

Recently, complementary precision dosing approaches that utilized model-informed initial dose selection (MIDS) have been proposed to support optimal initial dose selection [13,14,15] and supplement on-treatment dose modification strategies such as TDM and toxicity-guided dosing [16]. Two approaches may be applied to support MIDS: a top-down approach known as population pharmacokinetic (popPK) modelling, and a bottom-up approach known as a physiologically based pharmacokinetic (PBPK) modelling. With popPK modelling, non-linear mixed-effect models are used to describe variability in observed pharmacokinetic (PK) behaviour within a population based on covariates known to influence exposure; this approach may be used to predict future exposure by fitting limited a priori data. In PBPK modelling, physiological data for a population are combined with physiochemical and in vitro data for a drug under specific trial conditions to simulate exposure in a virtual population [17,18]. Simulated data may be compared to observed data from a matched population to define the performance of the PBPK model. Population PK models for epirubicin have been applied to describe the relation between epirubicin exposure and the incidence of haematologic toxicity [19], and to associate routinely collected demographic characteristics with epirubicin exposure [5,20].

The development of a PBPK model for epirubicin provides the capacity to (i) define the impact of additional molecular and physiological characteristics that are not routinely collected on epirubicin exposure, (ii) simulate exposure in populations that have not been studied in clinical trials (e.g., different races, age groups, etc.), and (iii) define the likely impact of pharmacogenetic variability on epirubicin exposure. Epirubicin is predominantly cleared by the liver, with renal elimination accounting for 20 to 25% of the dose. The reported primary enzyme involved in the hepatic clearance of epirubicin is UDP-glucuronosyltransferase (UGT) 2B7 [21]. In this regard, reduced UGT2B7 protein expression and/or activity caused by single nucleotide polymorphisms (SNPs) in the *UGT2B7* gene has been associated with increased epirubicin exposure and reduced metabolic clearance [21]. The most notable example of pharmacogenomic-guided epirubicin dosing involved the *UGT2B7 -161C>T* SNP. This SNP has been associated with a reduction in epirubicin clearance and increased AUC [22,23]; importantly, this SNP has also been demonstrated to predict grade ¾ leucopenia in early breast cancer patients treated with adjuvant or neoadjuvant FEC100 (5-fluorouracil 500 mg/m^2^, Epirubicin 100 mg/m^2^ and cyclophosphamide 500 mg/m^2^). 

The primary objective of this study was to identify physiological and molecular characteristics driving variability in epirubicin AUC using PBPK modelling. Identification of these characteristics informs analyses of ‘exposure biomarkers’ for epirubicin that can be evaluated using routinely collected samples from randomised controlled trials and can facilitate non-invasive optimal initial dose selection for this drug [24,25]. The second objective of this study was to define the association between epirubicin plasma concentration and tissue concentrations, with a focus on tissues relevant to either the therapeutic efficacy (adipose/breast tissue), or the incidence of toxicity (cardiac, hepatic) for this drug.

## 2. Materials and Methods

### 2.1. Materials and Chemical Information

Epirubicin (hydrochloride) was purchased from Cayman Chemical (Ann Arbor, MI, USA). UDP-glucuronic acid (UDPGA; trisodium salt) was purchased from Sigma-Aldrich (St Louis, MO, USA). Fluconazole was obtained from Pfizer Australia (Sydney, NSW, Australia). Alamethicin (from *Trichoderma viridae*) was purchased from AG Scientific (San Diego, CA, USA). Solvents and other reagents used were of analytical reagent grade or higher.

### 2.2. Human Liver Microsomes

Pooled human liver microsomes (HLMs) were prepared by mixing equal amounts of protein from five human livers (H7, 44-year-old female; H10, 67-year-old female; H12, 66-year-old male; H29, 45-year-old male; and H40, 54-year-old female), obtained from the human liver bank of the Department of Clinical Pharmacology of Flinders University. Approval for the use of human liver tissue in xenobiotic metabolism studies was obtained from the Flinders Clinical Research Ethics Committee. HLMs were prepared by differential centrifugation, as described by Bowalgaha et al. [26]. Microsomes were activated by pre-incubating on ice for 30 min in the presence of alamethicin (50 μg/mg microsomal protein) prior to inclusion in the incubation matrix [27].

### 2.3. Epirubicin Glucuronidation Assay

Assay conditions for epirubicin glucuronidation by HLMs were optimised for protein concentration, incubation time and epirubicin concentration range [28,29]. Incubations in a total volume of 200 µL contained MgCl_2_ (4 mM), potassium phosphate (0.1 M; pH 7.4), epirubicin (in DMSO 2% *v*/*v*), activated HLMs (0.01 mg), and UDPGA (5 mM). A 5 min pre-incubation at 37 °C was performed to thermodynamically equilibrate the mixture; reactions were initiated by the addition of UDPGA. Reactions to form epirubicin glucuronide were performed over 120 min at 37 °C in a shaking water bath and were terminated by the addition of 400 µL of ice-cold methanol containing 0.1% formic acid. The reaction mix was centrifuged at 4000× *g* for 10 min at 10 °C and a 300 µL aliquot of the supernatant fraction was transferred into LC-MS vials. Microsomal incubations were performed in the presence of fluconazole (10–2500 µM) to define the contribution of microsomal UGT2B7 to epirubicin glucuronidation by HLMs.

### 2.4. Quantification of Epirubicin Glucuronide Formation

Epirubicin glucuronide formation was quantified by liquid chromatography mass spectrometry (LC-MS) performed on an Agilent 1290 infinity liquid chromatography (LC) system coupled to an Agilent 6495B triple-quadrupole mass spectrometer (MS; Agilent Technologies, Santa Clara, CA, USA) fitted with a Zorbax Eclipse Plus C18 analytical column (1.8 µM, 2.1 mm × 50 mm; Agilent, Santa Clara, CA, USA). Epirubicin glucuronide was separated from the incubation matrix by using a mobile phase comprising 28% acetonitrile and 0.1% formic acid in water at a flow rate of 0.2 mL/min. Control incubations in the absence of the cofactor (UDPGA), substrate (epirubicin), and microsomal protein were analysed in parallel to incubation samples to confirm correct product detection. 

The MS source parameters were as follows: sheath gas flow rate of 11 L/min, gas flow rate of 14 L/min, gas temperature of 200 °C, nebulizer pressure of 35 psi and capillary voltage of 1500 V. Multiple reaction monitoring (MRM) was used to monitor the precursor transition ion at 720.22 m/z, with the optimised conditions around the product ions listed in Table 1. Epirubicin glucuronide was eluted at a retention time of 1.6 min.

### 2.5. Data Analysis (In Vitro Kinetics)

The kinetics of microsomal epirubicin glucuronidation (Michaelis constant, Km and maximal reaction velocity, Vmax) were determined by fitting experimental data using the Michaelis–Menten equation in GraphPad Prism 9.3.1 (San Diego, CA, USA). Fluconazole inhibition of microsomal epirubicin glucuronidation was determined by fitting experimental data to the competitive inhibition model using GraphPad Prism 9.3.1 (San Diego, CA, USA). In vitro kinetic data (Km and Vmax) generated in these microsomal incubations were used as input parameters in the PBPK model to describe epirubicin clearance by UGT2B7. 

### 2.6. Development and Verification of Epirubicin PBPK Model

#### 2.6.1. Structural Model

A full-body PBPK model to simulate the concentration time profile for epirubicin following a single IV dose infused over 3 min was developed using Simcyp^®^ version 19.1 (Certara, Sheffield, UK). The differential equations utilised by Simcyp to construct the PBPK model from physiochemical and in-vitro data have been described previously by [30].

#### 2.6.2. Development of Epirubicin Compound Profile

The physiochemical, blood binding, distribution and elimination parameters for epirubicin, along with parameters defining induction of UGT2B7 by epirubicin are summarized in Table 2. Physiochemical parameters were based on published literature values [31], unless specified blood binding and distribution parameters were predicted by the model based on the physiochemical parameters of the drug using in-built functions within the Simcyp simulator. Microsomal clearance data (assigned to UGT2B7 based on fluconazole inhibition) were based on in vitro incubations (see Methods section). Renal clearance (CL_R_) was calculated based on published clearance values [6]. Induction parameters for UGT2B7 were defined based on LC-MS proteomic data using HepG2 cells generated in this laboratory. 

#### 2.6.3. Population Profile

The epirubicin compound profile was built and verified using the Sim-Cancer population profile. The Sim-Cancer population profile was also used in simulations to characterise the association between epirubicin plasma and tissue concentrations, and to characterise the physiological and molecular parameters associated with variability in epirubicin exposure.

#### 2.6.4. Simulated Trial Design

For development of the epirubicin profile, simulations comprised 90 subjects divided across 10 trials with 9 subjects in each trial. During verification of the epirubicin compound profile, simulations were performed in 10 trials comprising age-, sex-, and ethnicity-matched subjects according to the protocol for the observed trial (dosing regimen and number of subjects) described in the following section. Parameters describing epirubicin exposure were assessed over 24 h following a single dose at 9:00 a.m. on day 1.

#### 2.6.5. Observed Clinical Data and Compound File Verification

Observed epirubicin pharmacokinetics were obtained from values reported in the literature by Robert, Vrignaud [33]. Sixteen metastatic breast carcinoma patients were subjected to a phase III comparative randomised protocol to assess the pharmacokinetics of epirubicin and doxorubicin. The epirubicin group received a combinatorial treatment of epirubicin (50 mg/m^2^), 5-FU (500 mg/m^2^), and cyclophosphamide (500 mg/m^2^). The initial dose of epirubicin was used to study pharmacokinetics. Epirubicin was administered first, followed by administration of the remaining chemotherapies after 1–2 h. Therefore, additional treatment could impact epirubicin pharmacokinetics. Plasma samples were obtained after 5, 10, 20, and 40 min, and after 1, 2, 4, 8, 24, 32, and 48 h for HPLC analysis of the unchanged drug and metabolites. Raw data obtained from this trial were reproduced and plotted for evaluation of the simulated epirubicin compound file. 

The epirubicin compound file was further verified by evaluating the impact of UGT2B7 inhibition, which was achieved by simulating the effect of fluconazole coadministration, and by evaluating the impact of renal function (evaluated as glomerular filtration rate; GFR). 

### 2.7. Population Characteristics Associated with Variability in Epirubicin Exposure 

The verified epirubicin profile was used to evaluate associations between physiological and molecular characteristics of the Sim-Cancer population and the logarithmically transformed epirubicin AUC (LnAUC). Ten trials from the Sim-Cancer population, each comprising 200 subjects, were simulated over 158 h, with a single 120 mg/m^2^ epirubicin dosed IV in a fasted state.

Univariate (simple) linear regression was performed using GraphPad Prism 9.3.1 (San Diego, CA, USA). Stepwise multivariate linear regression analysis was performed using IBM^®^ SPSS^®^ Statistics 26 (New York, NY, USA). Linear regression was used to evaluate associations between the physiological and molecular characteristics identified in Appendix A and epirubicin LnAUC. Continuous variables were evaluated for normality and non-linearity of association; binary characteristics (sex) were coded as nominal variables. A multivariable linear regression model was developed by stepwise forward inclusion of significant characteristics identified in the univariable regression analysis based on improvement in model R^2^. Back transformation of the model-predicted logarithmically transformed AUC was performed to plot correlations between the simulated and model-predicted AUC.

## 3. Results

### 3.1. Characterisation of In Vitro Epirubicin Glucuronidation

Epirubicin glucuronide formation by HLM was best described by a single-enzyme Michaelis–Menten equation (Figure 1A). The kinetic parameters derived for epirubicin glucuronidation were a K_m_ of 26.2 ± 5.48 µM and V_max_ of 2896.6 ± 212.8 AU. The selective UGT2B7 inhibitor fluconazole was included in incubations (final concentration 100 to 2500 µM) to confirm the involvement of UGT2B7 in human liver microsomal epirubicin glucuronidation. The IC_50_ for fluconazole inhibition of epirubicin glucuronidation by HLMs was 770.7 ± 158.1 µM, with a maximal observed inhibition of 75% (Figure 1B). These data support UGT2B7 as the major enzyme involved in human liver microsomal epirubicin glucuronidation.

### 3.2. Verification of the Epirubicin PBPK Model

The accuracy of the epirubicin compound profile was assessed using an age-, sex-, and race-matched single-dose trial [34]. Ten simulated trials were performed with epirubicin administered IV at a dose of 50 mg/m^2^ in trials comprising 9 female subjects aged between 20 and 50 years. Epirubicin plasma concentration, monitored over 48 h, was used to define the simulated epirubicin maximal concentration C_max_ and AUC. The mean (±SD) simulated AUC and C_max_ in the validation cohort were 1324 ± 20.0 ng/mL·h and 434 ± 42.6 ng/mL, respectively; these values are 1.1- and 1.6-fold higher than the respective measured parameters. The simulated mean (95% confidence interval; CI) and mean observed plasma concentration time profiles are shown in Figure 2. In all cases, the mean simulated epirubicin plasma concentration at each measured time point was within 1.6-fold of the respective observed plasma concentration.

Consistent with the reported importance of UGT2B7 in epirubicin metabolism in vivo, coadministration of steady-state fluconazole (200 mg daily for 7 days) resulted in a 54% increase in the single-dose epirubicin AUC. Notably, as an intravenously administered drug, the C_max_ for epirubicin was only modestly impacted (increased by 5%).

### 3.3. Epirubicin Exposure in Oncology Cohort

The mean, standard deviation (SD) and range of epirubicin AUC and C_max_ values describing exposure to epirubicin in a cohort of 200 oncology patients are reported in Table 3. Marked variability in epirubicin AUC and C_max_ was observed; by way of example, AUC values ranged from 2980 to 12,710 ng/mL·h (mean 5374 ng/mL·h). Simulated epirubicin AUC and C_max_ values, and the variability in these parameters, were consistent with observed exposure profiles [3].

### 3.4. Epirubicin Clearance Pathways

The mean contribution of renal elimination, defined by the fraction of epirubicin excreted unchanged in the urine (F_e_), was 20.95%. The mean contributions of hepatic and renal UGT2B7 to total epirubicin clearance were 75.3% and 4.3%, respectively (Figure 3).

### 3.5. Determination of Population Characteristics Affecting Epirubicin Clearance

The results of univariate linear regression analyses considering the association between molecular and physiological characteristics and epirubicin LnAUC are presented in Appendix A. Multivariable linear regression modelling by stepwise inclusion of parameters described in Appendix A identified hepatic UGT2B7 abundance, albumin concentration, age, renal UGT2B7 abundance, body surface area (BSA), glomerular filtration rate (GFR), haematocrit, and sex as the key covariates associated with variability in epirubicin LnAUC. By accounting for these factors, it was possible to explain 87% of the variability in epirubicin LnAUC within the oncology population (Figure 4, Table 4 and Table 5). The single most important covariate associated with variability in epirubicin LnAUC was hepatic UGT2B7 abundance; accounting for this covariate alone explained 56% of the variability in epirubicin LnAUC. Furthermore, exclusion of body surface area in the model led to a minor decrease in predicted variability to 83% (Appendix A).

### 3.6. Associations between Epirubicin Plasma and Tissue Concentration

The concordance between simulated epirubicin plasma and individual tissue concentrations are shown in Figure 5. Except for epirubicin concentration in the brain (R^2^ = 0.56), there was limited concordance between epirubicin tissue and plasma concentrations (R^2^ < 0.22). Notably, while the highest mean tissue AUC was observed in skeletal muscle (169,241 ng/mL·h), the comparatively slow distribution into this tissue resulted in a markedly lower C_max_ compared to other tissues. Indeed, despite comparable AUCs, the mean epirubicin C_max_ in cardiac tissue (31,952 ng/mL) was >10-fold higher than the epirubicin Cmax in skeletal muscle (2868 ng/mL) (Table 6). 

## 4. Discussion

The present study describes the development and evaluation of a full-body PBPK model to assess systemic and individual organ exposure to epirubicin. Multi-variable linear regression modelling demonstrated that variability in simulated systemic epirubicin exposure following intravenous injection was primarily driven by differences in hepatic and renal UGT2B7 expression, plasma albumin concentration, age, BSA, GFR, haematocrit and sex. By accounting for these factors, it was possible to explain 87% of the variability in epirubicin in a simulated cohort of 2000 oncology patients aged between 20 and 95 years. The single most important factor in defining simulated systemic epirubicin exposure was hepatic UGT2B7 expression, which alone accounted for 56% of variability in exposure within the simulated cohort.

Current dosing guidelines for epirubicin account for age, BSA, renal function and sex. Typically, epirubicin doses are reduced in patients with a serum creatinine > 5 mg/dL. Epirubicin is quite well tolerated in patients with chronic renal failure undergoing haemodialysis [35]. Dose reductions in elderly patients are also well tolerated [36]. Data regarding the value of BSA-guided epirubicin dosing are contentious, with multiple studies suggesting that BSA-guided dosing is of limited value [5,37]. In accordance with this, the data presented suggest that incorporating BSA-based dosing into the model only contributes to 4% of exposure variability. This highlights the importance of understanding other contributing variables in epirubicin dosage. Sexual dimorphism around DME expression could perhaps be driven by hormonal differences [38] and may influence epirubicin metabolism; however, further understanding of this is required. Reassuringly, the major factors currently accounted for when guiding epirubicin dosing are consistent with the physiological parameters identified in the multiple linear regression modelling performed in the current study and with prior non-linear mixed-effects modelling (NONMEM) analyses involving epirubicin. Wade, Kelman [4] demonstrated that by accounting for differences in sex and age it was possible to reduce unexplained variability in epirubicin clearance from 50 to 42%. Consistent with the major importance of UGT2B7 expression in defining epirubicin exposure, prior analyses have consistently demonstrated that a large proportion of the variability in epirubicin exposure cannot be accounted for based on routinely collected physiological parameters including age, sex, BSA and renal function. Currently, there is no reliable biomarker to define hepatic UGT2B7 expression in individual patients, and assessment of UGT2B7 genotype is of limited value [25]. However, in recent years liver-derived extracellular vesicles (EVs) have emerged as a potential universal ADME biomarker [39,40,41,42,43,44]. It is plausible that quantification of UGT2B7 expression in EVs may serve as a robust approach to estimate hepatic UGT2B7 expression in individual patients, thereby supporting dose individualisation for drugs such as epirubicin.

PBPK modelling and simulation is an established tool to support drug discovery and development and is a core element of the regulatory approval process in many jurisdictions [18]. Recent studies have further demonstrated the potential role of PBPK in predicting covariates affecting variability in drug exposure resulting from differences in patient characteristics [14,15,45], giving rise to the intriguing potential for this platform to support model-informed precision dosing [46,47]. This model provides an important foundation for establishing a PK/PD relationship for epirubicin; however, further work on population-based dose predictive modelling is imperative to inform optimal therapeutic windows for individualized dosage. The major limitation of the current study is the lack of observed clinical data to support the validation of the regression models. Currently, these models are based on a mechanistic systems pharmacology understanding and would require confirmation with in vivo clinical data to warrant implementation. A second limitation of the current study is the lack of observed tissue concentration measurements to support the lack of concordance between plasma concentration and tissue concentrations. While the overall simulated volume of distribution (25.265 L/kg) for epirubicin is consistent with reported in vivo data [6], the specific tissue distribution for this drug in vivo has not been reported.

Simulated epirubicin clearance was consistent with clinical observations of epirubicin plasma clearance as studied by Robert [34]. Notably, there was limited concordance between systemic epirubicin exposure and the exposure of individual organs to epirubicin. Except for epirubicin concentration in the brain (R^2^ = 0.56), there was limited concordance between tissue and systemic (plasma) epirubicin concentrations (R^2^ < 0.22). The highest mean tissue AUC was observed in skeletal muscle (169,241 ng/mL·h); however, the comparatively slow distribution into this tissue resulted in a markedly lower C_max_ compared to other tissues. Indeed, despite comparable AUCs, the mean epirubicin C_max_ in cardiac tissue (31,952 ng/mL) was >10-fold higher than the epirubicin C_max_ in skeletal muscle (2868 ng/mL). The extensive distribution of epirubicin into cardiac tissue is consistent with the well-established cardiac toxicity profile for this drug [10,48]. The limited concordance between plasma and cardiac epirubicin concentrations (r^2^ = 0.2188) indicates that evaluation of plasma epirubicin concentration is unlikely to be useful in identifying patients at greatest risk of suffering cardiac toxicity when administered epirubicin.

## Figures and Tables

**Figure 1 pharmaceutics-15-01222-f001:**
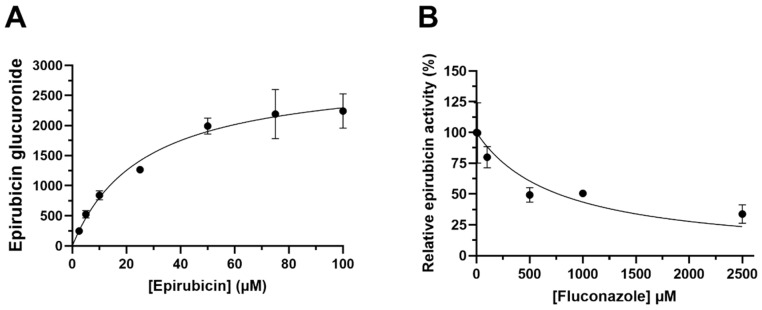
(**A**) Michaelis–Menten Kinetics of epirubicin by formation of epirubicin glucuronide in HLMs (R^2^ = 0.96). Pooled HLMs (2 mg/mL) were incubated for 2 h with incremental amounts of epirubicin (between 2.5–100 µM) and epirubicin glucuronide was detected in the absence of standards. (**B**) Normalised activity of epirubicin by inhibition of epirubicin glucuronide formation in HLMs. HLMs were incubated for 2 h with 25 µM epirubicin and increasing amounts of fluconazole (between 10–2500 µM). Epirubicin glucuronide was detected, and response was measured relative to the control in the absence of fluconazole. Mean peak area response ± S.D. is measured in arbitrary units. Data were generated in duplicate with standard deviation displayed by error bars.

**Figure 2 pharmaceutics-15-01222-f002:**
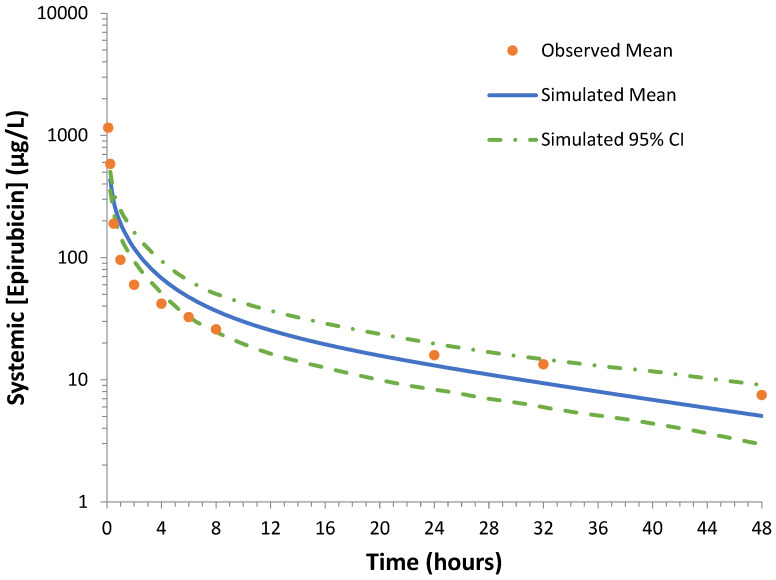
Representative overlay showing the simulated and observed epirubicin plasma concentration time curves over 48 h following a single oral dose (50 mg/m^2^). The solid blue line represents the mean simulated epirubicin plasma concentration, the dotted green lines represent the 95% confidence interval (CI) for the simulated data and the orange dots represent the mean observed data [34]. No data regarding variability in observed data was reported in the original publication.

**Figure 3 pharmaceutics-15-01222-f003:**
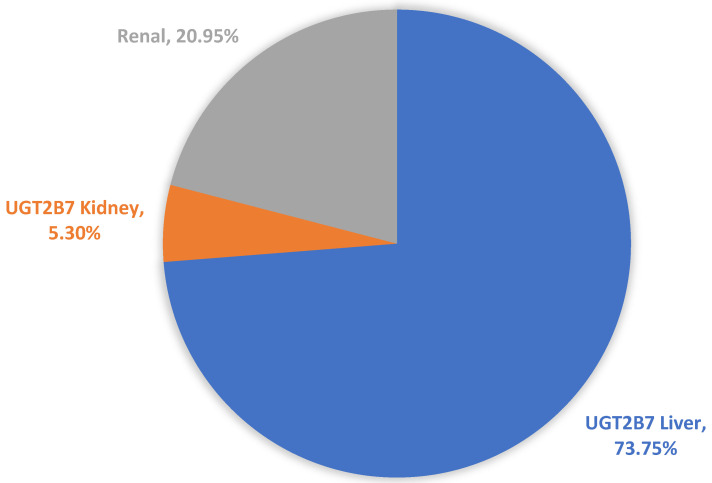
A representative pie-chart showing the relative contribution (geometric mean %) of the f_m_ (hepatic and renal epirubicin) and the f_e_ unchanged by renal clearance.

**Figure 4 pharmaceutics-15-01222-f004:**
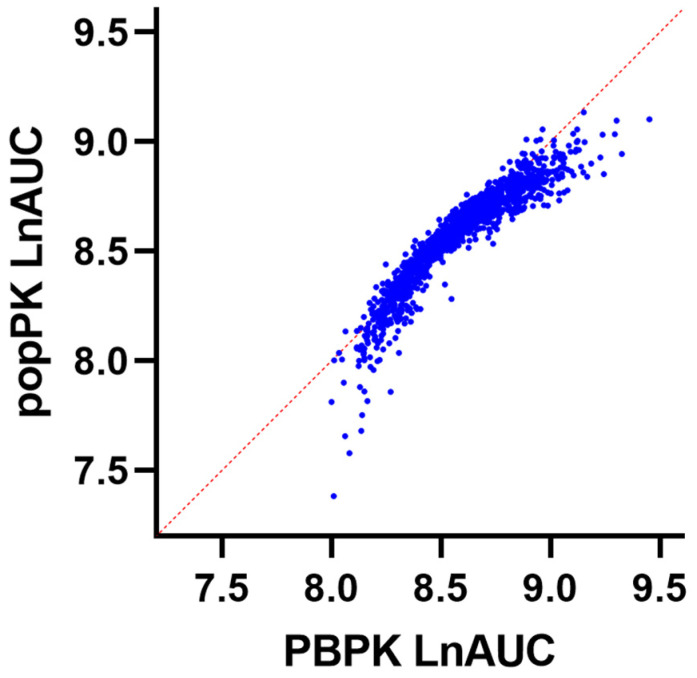
Multivariate linear regression analysis showing the correlated relationship between the simulated popPK natural log-transformed AUC (LnAUC) and simulated PBPK LnAUC.

**Figure 5 pharmaceutics-15-01222-f005:**
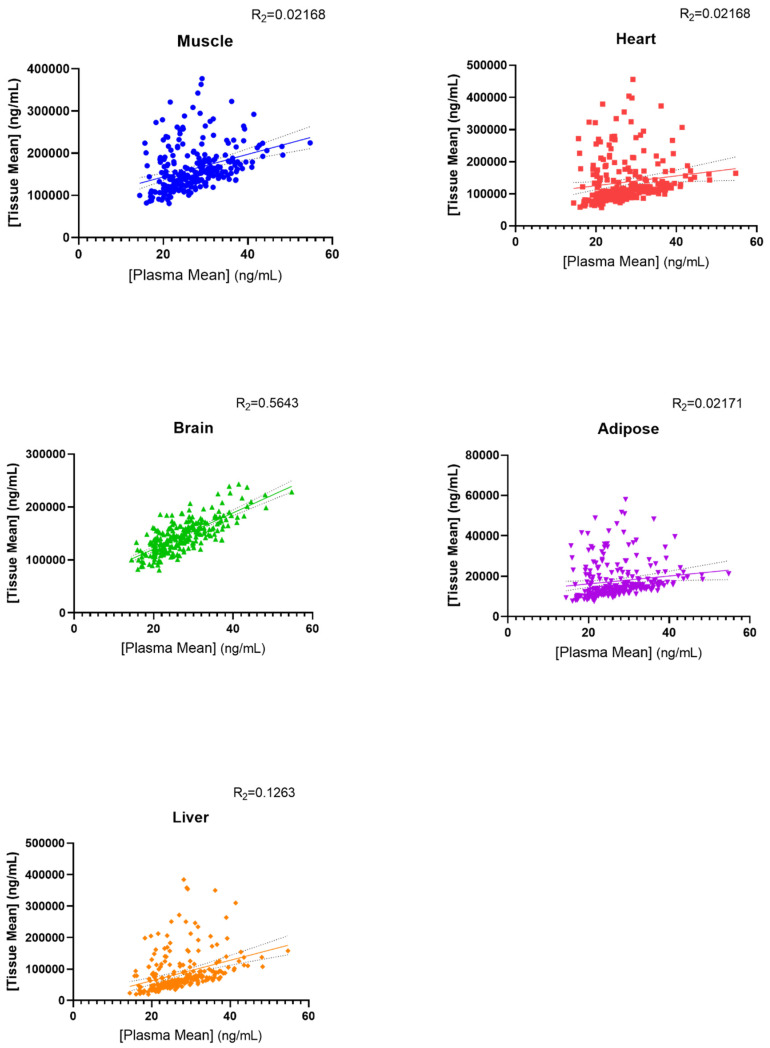
Linear regression analysis evaluating the relationship between simulated maximum epirubicin tissue and plasma concentrations in a Sim-Cancer cohort following a single 50 mg/m^2^ dose.

**Table 1 pharmaceutics-15-01222-t001:** Multiple Reaction Monitoring (MRM) scan parameters for the optimised epirubicin glucuronide ions.

Precursor Ion (m/z)	Product Ion (m/z)	Dwell (s)	Fragmentor (V)	Collision Energy (V)	Cell Acceleration (V)	Polarity
720.22	702.2	200	380	15	4	Positive
720.22	361.2	200	380	36	4	Positive
720.22	324.2	200	380	20	4	Positive
720.22	306	200	380	16	4	Positive

**Table 2 pharmaceutics-15-01222-t002:** Compound profile for epirubicin based on the physiochemical properties detailed.

Phys Chem	
Molecular Weight (g/mol)	543.52
log Po:w	1.41
Species	Diprotic Base
pKa (Strongest Acidic)	8.010
pKa 2 (Strongest Basic)	10.030
**Blood Binding**	
B/P	0.729
f_u_	0.23
**Distribution (full PB-PK model)**	
Vss (L/Kg)	25.265
Prediction Method	Rogers and Rowland [32]
Kp Scalar	25
**Elimination**	
HLM—UGT2B7 (Km; µM)	26.2
HLM—UGT2B7 (Vmax; pmol/min/mg protein)	2897
HLM—UGT2B7 (fu)	1
Additional clearance—CL_R_ (L/h)	9.0
**Interaction**	
UGT2B7 (IndC50; µM)	0.368
UGT2B7 (Indmax)	13.95

Po:w, neutral species octanol: water partition coefficient; B/P, blood-to-plasma partition ratio; fu, fraction unbound; Vss, steady state volume of distribution; Kp scalar: scalar applied to all predicted tissue partition values; CLR, renal clearance; IndC50, inducer concentration to achieve half-maximal induction; IndMax, maximal fold induction. Notes: Prediction of tissue distribution based on the model of Rogers and Rowland.

**Table 3 pharmaceutics-15-01222-t003:** Descriptive statistics showing the mean, variance and range of epirubicin exposure in the simulated oncology cohort.

Statistic	AUC (ng/mL·h)	CMax (ng/mL)	Dose (mg)	CL (Dose/AUC) (L/h)
Mean	5374	12,683	213.2	41.6
Median	5197	12,653	212.1	40.5
Geometric Mean	5252	12,654	211.8	40.3
90% confidence interval (lower limit)	5211	12,622	211.0	40.0
90% confidence interval (upper limit)	5293	12,686	212.7	40.7
5th centile	3776	11,296	176.4	26.7
95th centile	7537	14,142	253.7	59.8
Skewness	0.99	0.12	0.29	0.50
cv	0.22	0.07	0.11	0.24
Min Val	2980	9678	149.8	14.8
Max Val	12,710	15,469	302.5	80.9
Fold	4.27	1.60	2.02	5.45
Std Dev	1191	864	24.1	10.2

**Table 4 pharmaceutics-15-01222-t004:** Multivariate linear regression analysis of the model-predicted variables affecting epirubicin LnAUC and their respective linearity regarding LnAUC. R^2^ of the model is 0.8690.

Variable	Estimated Ln AUC (ng/mL·h)	Standard Error	Range (95% CI)	R^2^ with Other Variables	*p* Value
Intercept (constant)	8.211	0.02994	8.152 to 8.269		<0.0001
Sex	−0.03709	0.003901	−0.04474 to −0.02944	0.2216	<0.0001
Age	0.00251	0.000174	0.002169 to 0.002850	0.5024	<0.0001
BSA	0.2669	0.01132	0.2447 to 0.2891	0.4266	<0.0001
Haematocrit	−0.00538	0.000373	−0.006110 to −0.004649	0.005414	<0.0001
Albumin	0.0111	0.000251	0.01060 to 0.01159	0.01572	<0.0001
GFR	−0.00178	0.000106	−0.001983 to −0.001568	0.5261	<0.0001
Liver UGT2B7	−9.77 × 10^−8^	1.34 × 10^−9^	−1.00× 10^−7^ to −9.51 × 10^−8^	0.2798	<0.0001
Kidney UGT2B7	−3.24 × 10^−7^	1.07 × 10^−8^	−3.45 × 10^−7^ to −3.03 × 10^−7^	0.03006	<0.0001

Model variables; sex 0 = Female, 1 = Male, age (years), body surface area (BSA) (m^2^), haematocrit (%), albumin (g/L), glomerular filtration rate (GFR) (mL/min/1.73 m^2^), liver UGT2B7 (pmol), kidney UGT2B7 (pmol).

**Table 5 pharmaceutics-15-01222-t005:** Stepwise multivariate linear regression analysis of predictors of epirubicin LnAUC by sequential addition according to best fit. Cumulative R^2^ of the final model (h) incorporating all predictors = 0.8690.

Model	R	R^2^	Adjusted R^2^	Std. Error of the Estimate	R^2^ Change
a	0.749 ^a^	0.561	0.561	0.141	0.561
b	0.821 ^b^	0.674	0.674	0.121	0.113
c	0.861 ^c^	0.741	0.740	0.108	0.067
d	0.886 ^d^	0.785	0.784	0.099	0.044
e	0.911 ^e^	0.830	0.830	0.087	0.046
f	0.922 ^f^	0.849	0.849	0.082	0.019
g	0.929 ^g^	0.863	0.863	0.079	0.014
h	0.932 ^h^	0.869	0.868	0.077	0.006

Model predictors; ^(a)^ LiverUGT2B7; ^(b)^ LiverUGT2B7, Albumin; ^(c)^ LiverUGT2B7, Albumin, Age; ^(d)^ LiverUGT2B7, Albumin, Age, KidneyUGT2B7; ^(e)^ LiverUGT2B7, Albumin, Age, KidneyUGT2B7, BSA; ^(f)^ LiverUGT2B7, Albumin, Age, KidneyUGT2B7, BSA, GFR; ^(g)^ LiverUGT2B7, Albumin, Age, KidneyUGT2B7, BSA, GFR, Haematocrit; ^(h)^ LiverUGT2B7, Albumin, Age, KidneyUGT2B7, BSA, GFR, Haematocrit, Sex.

**Table 6 pharmaceutics-15-01222-t006:** PBPK-predicted epirubicin mean Cmax (ng/mL) and mean AUC (ng/mL·h) in tissue and plasma over 168 h after IV injection in a Sim-Cancer population.

Tissue	Mean Cmax (ng/mL)	Mean AUC (ng/mL·h)
Plasma	979	4530
Muscle	2868	169,241
Heart	31,952	144,482
Brain	22,410	147,660
Adipose	1049	18,680
Liver	13,973	92,446

## Data Availability

Data supporting the findings of this study are available from the corresponding author upon reasonable request.

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
