# Peer review of "A Physiologically Based Pharmacokinetic Model to Predict Determinants of Variability in Epirubicin Exposure and Tissue Distribution"

_pharmaceutics, 2023, doi:10.3390/pharmaceutics15041222_

Round 1

Reviewer 1 Report

Dr Ansaar and colleagues built a PBPK model based on in vitro data included in a generalized cancer PBPK model in Simcyp and evaluated  the model by comparing simulations with summarized observed plasma PK data. The manuscript focuses on predicting variability in PK among patients based on differences in liver and renal function, and shows interesting (unaccounted) variability in tissue accumulation of epirubicin. This is of interest to readers of pharmaceutics as epirubicin has a small therapeutic window and the current practice of BSA-based dosing leads to toxicity or reduced efficacy in some patients. However, the results of the model were not validated using observed variability in clinical PK-data nor by comparing PK with PD. Therefore, I have the following suggestions to evaluate the predictive value of the model:

Can the authors consider validating the most predictive covariables, eg by by comparing the results to observed plasma PK data at the extremes of eGFR, liver function and with or without concomitant fluconazole administration?

In the introduction the authors describe “It is possible to utilise therapeutic drug monitoring (TDM) to guide epirubicin dosing”  However, as no clear exposure response relationship has been established and target window has been defined, this is not correct. Can the authors rephrase this sentence. Can the authors make clear that for model-informed dose selection a target window is needed and can they add the lack of a clear PKPD relationship as a limitation to the application of the model in the discussion?

As epirubicin dosing is based on BSA, can the authors consider comparing the results of the final model with a model without BSA and thereby show how much of the variability can be explained by other parameters then BSA. Then can the authors consider discussing whether individualized epirubicin dosing based on other characteristics than BSA would benefit some patients?  

The method and result section of the abstract should include the in vitro experiments that were performed to inform the model and make clear that covariates based PK profiles were generated using simulations only.

Can the authors add a paragraph in the method section that describes which clinical data were from literature used (how many patients, doses and whether PK samples were measured after a single dose or at steady state and whether raw data or summarized data from the publication were used) to evaluate the model.

Table 2 should be explained further. Eg for a reader not familiar with SimCyp, it is unclear what Prediction Method =2 means.

Figure 2 does shows observed vs predicted PK data. Can the authors describe in the legend that a summary of the observed PK data of Robert et al are presented (not clearance values). As the manuscript focuses on predicting variability, can the authors consider including confidence intervals of the observed data?

Instead of showing simulated plasma to tissue concentrations in figure 5 it would be helpful to show observed versus predicted tissue concentrations of preclinical or clinical data. Currently the dots and trendline are misleading as the dots donot present observed but simulated data.

Can the authors describe study limitations including the current lack of robust clinical validation of the regression models and tissue concentration measurements should be included in the discussion.

Author Response

Please see attached responses

Reviewer 2 Report

The topic of the manuscript is interesting and fits well the scope of Pharmaceutics. The reviewer feels it can be accepted after some minor amendments.

(1) The hepatic microsomes were only prepared from  5  donors . How to assure this model is reliable and represent the general population? 

(2) How to measure the glucuronide metabolite? Did the authors have authentic standard? 

(3) The authors should briefly discuss the metabolic / elimination pathways of Epirubicin in introduction.

Author Response

Please see attached responses

Round 2

Reviewer 1 Report

The questions were adequately adressed by the authors and the manuscript has improved. I have no further suggestions.